# Hyperspectral Image Enhancement by Two Dimensional Quaternion Valued Singular Spectrum Analysis for Object Recognition

Yuxin Lin [1], Bingo Wing-Kuen Ling [1,*], Lingyue Hu [1], Yiting Zheng [2], Nuo Xu [1], Xueling Zhou [1] and Xinpeng Wang [1]

1   School of Information Engineering, Guangdong University of Technology, Guangzhou 510006, China; a19860075121@163.com (Y.L.); hulingyue1005@163.com (L.H.); 13426910493@163.com (N.X.); zzhouxueling@gmail.com (X.Z.); wxp@gpnu.edu.cn (X.W.)
2   Faculty of Physical Sciences and Engineering, Cardiff University, Cardiff, Wales CF10 3AT, UK; zhengyiting1995@163.com
*   Correspondence: yongquanling@gdut.edu.cn

**Abstract:** This paper proposes a two dimensional quaternion valued singular spectrum analysis based method for enhancing the hyperspectral image. Here, the enhancement is for performing the object recognition, but neither for improving the visual quality nor suppressing the artifacts. In particular, the two dimensional quaternion valued singular spectrum analysis components are selected in such a way that the ratio of the interclass separation to the intraclass separation of the pixel vectors is maximized. Next, the support vector machine is employed for performing the object recognition. Compared to the conventional two dimensional real valued singular spectrum analysis based method where only the pixels in a color plane is exploited, the two dimensional quaternion valued singular spectrum analysis based method fuses four color planes together for performing the enhancement. Hence, both the spatial information among the pixels in the same color plane and the spectral information among various color planes are exploited. The computer numerical simulation results show that the overall classification accuracy based on our proposed method is higher than the two dimensional real valued singular spectrum analysis based method, the three dimensional singular spectrum analysis based method, the multivariate two dimensional singular spectrum analysis based method, the median filtering based method, the principal component analysis based method, the Tucker decomposition based method and the hybrid spectral convolutional neural network (hybrid SN) based method.

**Keywords:** two dimensional quaternion valued singular spectrum analysis; multichannel two dimensional signals; hyperspectral image; object recognition; component selection



## 1. Introduction

With the development of the terrain survey, recognizing different objects on the wide ground becomes more and more important for the agriculture monitoring and the forestry management [1]. However, it requires a lot of instruments and labor resources for identifying and surveying the ground facilities because of the complicated and changeable ground environments.

In order to address the above difficulty, the remote sensing approach is employed. In particular, the hyperspectral image is obtained by the sensor [2]. Here, the hyperspectral image is an image having many color planes. Each pixel vector in the hyperspectral image belongs to a type of object such as the soil, the water and the vegetation [3]. Hence, identifying and surveying the ground facilities become the object recognition of the pixel vectors in the hyperspectral image. Recently, this object recognition problem has received a lot of attentions in the remote sensing community [4,5].

There are two main types of preprocessing approaches for enhancing the hyperspectral image for performing the object recognition. The first type of the approaches is the pixel vector based methods [6–8]. The common pixel vector based method is the principal component analysis based method. However, the correlation among the pixels within each color plane is ignored. The second type of the approaches is the color plane based method [9]. The common color plane based method is the two dimensional real valued singular spectrum analysis based method. However, the correlation among different color planes is ignored. To address the above difficulties, a joint pixel domain and spectral domain technique called the tensor decomposition based method is proposed for performing the preprocessing of the hyperspectral image [10–12]. However, the required computational power of the tensor decomposition based method is very high. Besides, with the popularity of the application of the convolutional neural network (CNN), the CNN based methods are proposed for improving the classification accuracy of the pixel vectors in the hyperspectral image. This is also a joint pixel and spectral domain approach [13].

On the other hand, a quaternion valued number is a number having three imaginary valued components and one real valued component [14–17]. By grouping the color planes of a hyperspectral image into the groups with each group having four color planes, forming the quaternion valued matrices, applying the quaternion valued singular spectrum analysis to the quaternion valued matrices, selecting the two dimensional quaternion valued singular spectrum analysis components and reconstructing the hyperspectral image, the correlation among both the pixels within each color plane and the pixels across different color planes are exploited [18]. Hence, this paper employs the two dimensional quaternion valued singular spectrum analysis based method for enhancing the hyperspectral image for performing the object recognition [19].

It is worth noting that the singular spectrum analysis has a characteristic that the singular spectrum analysis components corresponding to the large singular values have the higher importance than those corresponding to the small singular values. This is because the singular spectrum analysis components corresponding to the small singular values usually refer to the noise components [20]. Hence, the conventional method for selecting the singular spectrum analysis components is to determine the singular spectrum analysis index such that the reconstructed signal is the sum from the singular spectrum analysis component with the largest singular value to the determined singular spectrum analysis component [21,22]. However, this approach does not help for performing the object recognition because the information for various classes of objects has not been exploited. On the other hand, the linear discriminant analysis is to maximize the interclass separation and minimize the intraclass separation of the feature vectors. It was shown to be effective for improving the classification accuracy in many pattern recognition applications [23–26]. Therefore, this paper proposes a linear discriminant analysis based method for selecting the two dimensional quaternion valued singular spectrum analysis components for enhancing the hyperspectral image for performing the object recognition.

It is worth noting that the selection of the singular spectrum analysis component does not involve the feature extraction as no feature is defined during the selection process. The main novelty and the major contributions of this paper are on the selection of the two dimensional quaternion valued singular spectrum analysis components for enhancing the hyperspectral image for performing the object recognition. Different from the dimension reduction based method such as the principal component analysis based method, the size of the hyperspectral image processed by our proposed method is the same as that of the unprocessed hyperspectral image. Additionally, our proposed method maximizes the ratio of the interclass separation to the intraclass separation of the pixel vectors. On the contrary, this is not the case for the dimension reduction based method. Second, by using the quaternion valued singular spectrum analysis, the correlation among different pixels within each color plane and that among different color planes are exploited. The outline of this paper is as follows. Section 2.1 reviews the basic operations of the two dimensional quaternion valued singular spectrum analysis. Section 2.2 presents our proposed method

for enhancing the hyperspectral image. Section 2.3 presents the datasets. Section 3 presents the computer numerical simulation results. Section 4 presents the discussions. Finally, the conclusion is drawn in Section 5.

## 2. Datasets and Our Proposed Methods

### 2.1. Review of the Two Dimensional Quaternion Valued Singular Spectrum Analysis

Since the quaternion valued signals consist of four components, the two dimensional quaternion valued singular spectrum analysis is effective for fusing the multichannel two dimensional signals together. The procedures for performing the two dimensional quaternion valued singular spectrum analysis are reviewed as follows:

### 2.1.1. Embedding Operation

Let $H^{a \times b}$ be the set of the $a \times b$ quaternion valued matrices. Let $x$ be a quaternion valued matrix with the size equal to $h \times w$ and $Z_{s,t}$ be the element in the $s^{\text{th}}$ row and the $t^{\text{th}}$ column of $x$. That is,

$$x = \begin{bmatrix} Z_{1,1} & Z_{1,2} & \dots & Z_{1,w} \\ Z_{2,1} & Z_{2,2} & \cdots & Z_{2,w} \\ \vdots & \vdots & \ddots & \vdots \\ Z_{h,1} & Z_{h,2} & \dots & Z_{h,w} \end{bmatrix} \in H^{h \times w} \tag{1}$$

It is worth noting that $x$ consists of four channels of matrices. Let the size of the window be $u \times v$. Here, $1 \leq u \leq h$ and $1 \leq v \leq w$. Similar to performing the embedding operation in the two dimensional real valued singular spectrum analysis, the window is moved from top to bottom and from left to right. Let the signal in the window with the starting index $(s, t)$ be $W_{s,t}$. That is,

$$W_{s,t} = \begin{pmatrix} Z_{s,t} & Z_{s,t+1} & \dots & Z_{s,t+v-1} \\ Z_{s+1,t} & Z_{s+1,t+1} & \cdots & Z_{s+1,t+v-1} \\ \vdots & \vdots & \ddots & \vdots \\ Z_{s+u-1,t} & Z_{s+u-1,t+1} & \dots & Z_{s+u-1,t+v-1} \end{pmatrix} \in H^{u \times v} \tag{2}$$

Then, the elements in $W_{s,t}$ are put into the column vector as follows:

$$\vec{W}_{s,t} = vec(W_{s,t}) = \begin{bmatrix} Z_{s,t} & \dots & Z_{s+u-1,t} & \dots & Z_{s,t+v-1} & \dots & Z_{s+u-1,t+v-1} \end{bmatrix}^T \in H^{uv \times 1} \tag{3}$$

Finally, the quaternion valued trajectory matrix is constructed as follows:

$$W = \begin{bmatrix} \vec{W}_{1,1} & \dots & \vec{W}_{1,w-v+1} & \dots & \vec{W}_{h-u+1,1} & \dots & \vec{W}_{h-u+1,w-v+1} \end{bmatrix} \in H^{uv \times (h-u+1)(w-v+1)} \tag{4}$$

### 2.1.2. Quaternion Valued Singular Value Decomposition

Let the involutions about the $i$ imaginary axis, the $j$ imaginary axis and the $k$ imaginary axis of $W$ be $W^i$, $W^j$ and $W^k$, respectively. Let the augmented quaternion valued trajectory matrix be

$$W^a = \begin{bmatrix} W^T & W^{iT} & W^{jT} & W^{kT} \end{bmatrix}^T \in H^{4uv \times (w-v+1)(h-u+1)} \tag{5}$$

Let $C^a$ be the covariance matrix of the augmented quaternion valued trajectory matrix. That is,

$$C^a = E\left(W^a W^{aH}\right) \tag{6}$$

To perform the two dimensional quaternion valued singular spectrum analysis, the quaternion valued singular value decomposition is performed on $C^a$. That is,

$$W^a = \sum_{p=1}^{r} W^a_p = \sum_{p=1}^{r} \sqrt{\lambda_p} u_p v_p^H \tag{7}$$

where $r$ is the maximum value of $p$ in $\{p : \lambda_p > 0\}$.

### 2.1.3. Reconstruction Stage

The grouping operation is to categorize the two dimensional quaternion valued singular spectrum analysis components into several groups and summing up all the two dimensional quaternion valued singular spectrum analysis components within the same group together. That is, the index set $\{1, 2, \ldots, r\}$ is partitioned into $M$ disjoint subsets. They are denoted as $I_1, I_2 \ldots, I_M$. Then, $W^a$ can be represented as

$$W^a = \hat{W}_1^a + \hat{W}_2^a + \cdots + \hat{W}_M^a \tag{8}$$

where $\hat{W}_m^a = \sum\limits_{p \in I_m} \sqrt{\lambda_p} u_p v_p^T$ for $k = 1, 2, \ldots, M$. It is worth noting that there are many different grouping methods. Therefore, this operation is not unique. Let $\hat{W}_{m,s,t}^a$ be the element in the $s^{\text{th}}$ row and the $t^{\text{th}}$ column of $\hat{W}_m^a$. Finally, let $WR_m$ be the first $uv$ rows of $\hat{W}_m^a$. That is,

$$WR_m = \begin{pmatrix} \hat{W}_{m,1,1}^a & \hat{W}_{m,1,2}^a & \cdots & \hat{W}_{m,1,(h-u+1)(w-v+1)}^a \\ \hat{W}_{m,2,1}^a & \hat{W}_{m,2,2}^a & \cdots & \hat{W}_{m,2,(h-u+1)(w-v+1)}^a \\ \vdots & \vdots & \ddots & \vdots \\ \hat{W}_{m,uv,1}^a & \hat{W}_{m,uv,2}^a & \cdots & \hat{W}_{m,uv,(h-u+1)(w-v+1)}^a \end{pmatrix} \tag{9}$$

Let $\widetilde{x}$ be the reconstructed quaternion valued matrix with the size equal to $h \times w$. It can be achieved by performing the averaging within each block, the averaging among the blocks and the de-Hankelization operation over the real valued component, the $i$ imaginary component, the $j$ imaginary component and the $k$ imaginary component of each $WR_m$ individually as that performed in the two dimensional real valued singular spectrum analysis.

It is worth noting that both the embedding and the de-Hankelization stages are the core operations in the singular spectrum analysis. It has been applied to many science and engineering applications successfully [27,28]. Moreover, the operation for generating the trajectory matrix in the singular spectrum analysis behaves similar to the convolution operation in the CNN. This is used to extract the local relationship among the pixels in the window. Besides, the quaternion singular value decomposition, as the extension of the singular value decomposition, has been successfully applied to many multichannel application, such as processing of a color image [29,30].

### 2.2. *Our Proposed Method*

It is worth noting that the feature vectors corresponding to various classes of objects are nonlinear separable. Therefore, the recognition accuracy is quite low. To address this problem, this paper proposes a two dimensional quaternion valued singular spectrum analysis based method for enhancing the hyperspectral image for performing the object recognition. In particular, the ratio of the interclass separation to the intraclass separation of the pixel vectors is maximized.

Our proposed algorithm consists of three stages. The first stage is to obtain the two dimensional quaternion valued singular spectrum analysis components of the groups of the hyperspectral image with four color planes. The second stage is to select the two dimensional quaternion valued singular spectrum analysis components. The third stage is to remove some imaginary parts of the two dimensional quaternion valued singular spectrum analysis components in the last group.

### 2.2.1. Obtaining the Two Dimensional Quaternion Valued Singular Spectrum Analysis Components of the Hyperspectral Image

All the color planes of the hyperspectral image are divided into the groups such that each group consists of four color planes. Then, the two dimensional quaternion valued

singular spectrum analysis is applied to each group of the color planes to obtain the corresponding components. Here, the window size of the two dimensional quaternion valued singular spectrum analysis is the same for all the groups of the color planes. Additionally, the color plane corresponding to the largest wavelength is set to the real part of the quaternion valued matrix, while the color planes corresponding to the second highest wavelength to the smallest wavelength are set to the $i$ imaginary part, the $j$ imaginary part and the $k$ imaginary part of the quaternion valued matrix, respectively. For the last group of the color planes, some zero color planes are inserted so that this group also contains four color planes. Here, the zero color planes are put starting from the $k$ imaginary part of the quaternion valued matrix.

Since the window size of the singular spectrum analysis plays an important role in the generation of the singular spectrum analysis components, it will affect the classification accuracy. In a hyperspectral image, the pixels in the same class are always gathered together. Hence, they have a closer relationship with one other. In order to exploit the local relationship among the pixels in the window, a window with a smaller size is preferred. Moreover, the larger the window size being selected will result to the higher computational power required. Therefore, this paper selects a window with a smaller size. In particular, the window size $3 \times 3$ is compared to the window size $5 \times 5$ in our proposed method and other singular spectrum analysis based methods for different datasets as shown in Table 1. Here, the datasets are the KSC dataset, the Indian pines dataset and the Botswana dataset. The details of these datasets will be discussed in Section 2.3. From Table 1, it can be seen that the classification accuracies achieved by our proposed method at the window size $5 \times 5$ perform better than those at $3 \times 3$ for all ratios of the total number of the pixel vectors in the training set to that in the dataset and for all the datasets. Likewise, the classification accuracies achieved by other singular spectrum analysis based methods at the window size $5 \times 5$ also perform better than those at $3 \times 3$ except for the two dimensional real valued singular spectrum analysis based method at the ratio of the total number of the pixel vectors in the training set to that in the dataset being equal to 10% testing on the KSC dataset. Therefore, the window size of the two dimensional quaternion valued singular spectrum analysis in this paper is chosen as $5 \times 5$.

**Table 1.** The comparison of our proposed method to other singular spectrum analysis based methods with different window sizes and different datasets.

| Methods | Window Sizes | The Overall Classification Accuracies Achieved by Our Proposed Method and Other Singular Spectrum Analysis Based Methods Using Different Ratios of the Total Number of the Pixel Vectors in the Training Set to That in the Dataset. | | | | | | | | |
|---|---|---|---|---|---|---|---|---|---|---|
| | | KSC | | | Indian | | | Botswana | | |
| | | 0.05 | 0.1 | 0.2 | 0.05 | 0.1 | 0.2 | 0.05 | 0.1 | 0.2 |
| Two dimensional quaternion valued singular spectrum analysis based method | $3 \times 3$ | 90.0323 | 93.0366 | 96.4320 | 91.3853 | 95.5313 | 97.0340 | 96.8952 | 98.9765 | 99.0699 |
| | $5 \times 5$ | 90.9604 | 94.8466 | 98.2519 | 94.6157 | 97.3960 | 98.3371 | 97.5097 | 99.0106 | 99.6933 |
| Two dimensional real valued singular spectrum analysis based method | $3 \times 3$ | 86.2591 | 92.3414 | 91.3541 | 89.1166 | 94.5824 | 97.7910 | 96.2241 | 97.3210 | 97.5813 |
| | $5 \times 5$ | 86.2793 | 90.0341 | 93.8937 | 90.8834 | 95.2775 | 98.4984 | 96.2807 | 98.0894 | 98.4279 |
| Multivariate two dimensional singular spectrum analysis based method | $3 \times 3$ | 88.3459 | 88.9645 | 92.4067 | 90.3336 | 95.1254 | 97.4640 | 93.6578 | 97.1158 | 98.0021 |
| | $5 \times 5$ | 89.5884 | 90.5273 | 96.2404 | 91.0612 | 95.8402 | 97.5552 | 96.8305 | 98.2259 | 99.2331 |
| Three dimensional real valued singular spectrum analysis based method | $3 \times 3 \times 3$ | 88.9831 | 92.2130 | 95.9770 | 91.8348 | 93.7769 | 97.1581 | 93.6016 | 97.7823 | 98.3652 |
| | $5 \times 5 \times 4$ | 89.3261 | 92.3765 | 97.2941 | 91.8453 | 94.3617 | 97.6297 | 94.5019 | 98.2600 | 99.0798 |

2.2.2. Selecting the Two Dimensional Quaternion Valued Singular Spectrum Analysis Components

For performing the object recognition of the hyperspectral image, different pixel vectors represent different objects. Let $X_{i,c}$ be the $i^{th}$ pixel vector of the reconstructed hyperspectral image corresponding to the $c^{th}$ class of objects. Let $N_c$ be the total number of the pixel vectors in the $c^{th}$ class of objects. Let $N$ be the total number of the classes of the objects in the reconstructed hyperspectral image. Let $\overline{X}_c$ be the mean of all the pixel vectors in the $c^{th}$ class of objects.

Since the two dimensional quaternion valued singular spectrum analysis components corresponding to the large singular values contain more information about the hyperspectral image while those corresponding to the small singular values contain less information about the hyperspectral image, only those corresponding to the large singular values are retained and considered for enhancing the hyperspectral image for performing the object recognition. However, selecting different two dimensional quaternion valued singular spectrum analysis components will yield different classification accuracies.

It is worth noting that the larger the ratio of the interclass separation to the intraclass separation of the pixel vectors will result to the more effective of the classification. Hence, this paper selects the two dimensional quaternion valued singular spectrum analysis components in such a way that the ratio of the interclass separation to the intraclass separation of the pixel vectors of the reconstructed hyperspectral image is maximized. Define the ratio of the interclass separation to the intraclass separation of the pixel vectors be:

$$
R = \frac{\frac{2}{N(N-1)} \sum\limits_{d=1}^{N} \sum\limits_{c=1}^{N} \left(\overline{X}_c - \overline{X}_d\right)^T \left(\overline{X}_c - \overline{X}_d\right)}{\frac{1}{N_c} \sum\limits_{c=1}^{N} \sum\limits_{i=1}^{N_c} \left(X_{i,c} - \overline{X}_c\right)^T \left(X_{i,c} - \overline{X}_c\right)}
\tag{10}
$$

Here, $\overline{X}_c$ is the mean of the vectors in the $c^{th}$ class of object and $X_{i,c}$ is the $i^{th}$ vector in the $c^{th}$ class of object.

First, all the pixel vectors are represented as the quaternion valued numbers. Second, the two dimensional quaternion valued singular spectrum analysis is applied to the quaternion valued matrix. Then, only the two dimensional quaternion valued singular spectrum analysis component with the largest singular value is selected to generate a reconstructed image. Next, all the pixel vectors of the reconstructed image are divided into the training set and the test set. The ratio of the interclass separation to the intraclass separation of the pixel vectors of the reconstructed image in the training set is computed. Third, the two dimensional quaternion valued singular spectrum component with the next largest singular value is added to the reconstructed image to generate an updated reconstructed image. Then, the ratio of the interclass separation to the intraclass separation of the pixel vectors of the undated reconstructed image in the training set is computed. This procedure is repeated. Finally, the two dimensional quaternion valued singular spectrum analysis components are selected in such a way that the ratio of the interclass separation to the intraclass separation of the pixel vectors of the corresponding reconstructed image in the training set is maximized. Figures 1–3 show the relationship between the total numbers of the selected two dimensional quaternion valued singular spectrum analysis components and the values of $R$ for different ratios of the total number of the pixel vectors in the training set to that in the whole hyperspectral image for the KSC image, the Botswana image and the Indian image, respectively. In particular, the ratios of the total number of the pixel vectors in the training set to that in the whole hyperspectral image are chosen as 0.05, 0.1 and 0.2 for the above three images.

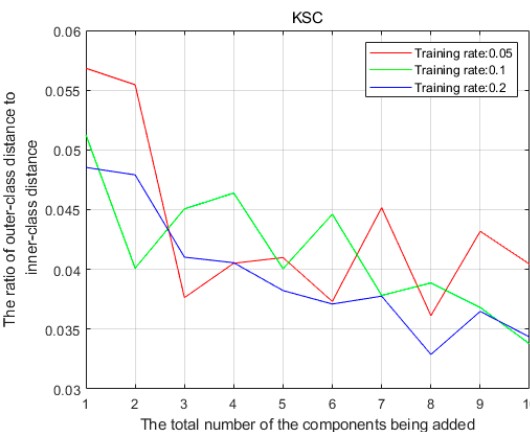

**Figure 1.** The ratios of the interclass separation to the intraclass separation of the pixel vectors at different total numbers of the selected two dimensional quaternion valued singular spectrum analysis components for the KSC image.

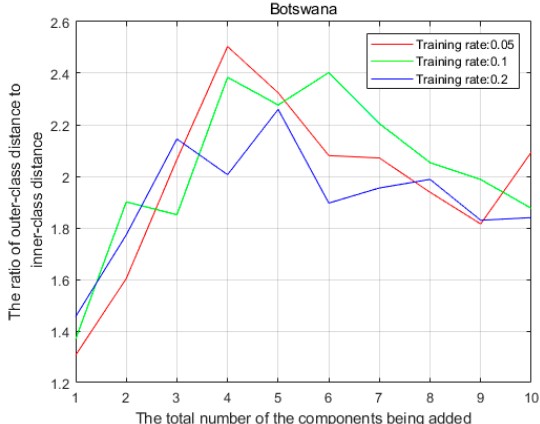

**Figure 2.** The ratios of the interclass separation to the intraclass separation of the pixel vectors at different total numbers of the selected two dimensional quaternion valued singular spectrum analysis components for the Botswana image.

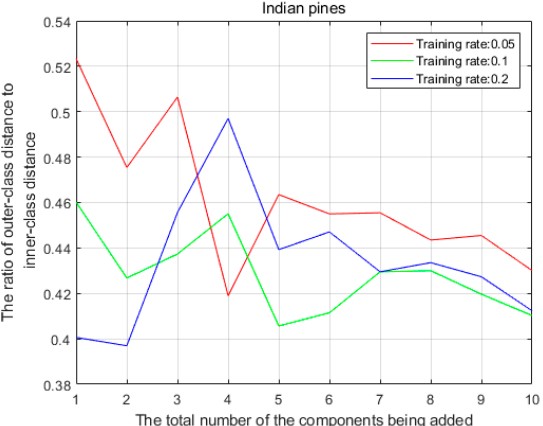

**Figure 3.** The ratios of the interclass separation to the intraclass separation of the pixel vectors at different total numbers of the selected two dimensional quaternion valued singular spectrum analysis components for the Indian pines image.

Since the window size of the two dimensional quaternion valued singular spectrum analysis is chosen as $5 \times 5$, the total number of the two dimensional quaternion valued

singular spectrum analysis components is equal to $4 \times 5 \times 5$. It can be seen from Figure 1 that the maximum value of $R$ occurs when only one two dimensional quaternion valued singular spectrum analysis component is selected for the KSC image. Therefore, the first two dimensional quaternion valued singular spectrum analysis component is selected for all the pixel vectors including those in the test set. On the other hand, for the Botswana image shown in Figure 2, the maximum values of $R$ occur when the first four, the first six and the first five two dimensional quaternion valued singular spectrum analysis components are selected at the ratios of the total numbers of the pixel vectors in the training set to that in the whole hyperspectral image being equal to 0.05, 0.1 and 0.2, respectively. In this case, the first four, the first six and the first five two dimensional quaternion valued singular spectrum analysis components are selected for all the pixel vectors including those in the test set at the ratios of the total numbers of the pixel vectors in the training set to that in the whole hyperspectral image being equal to 0.05, 0.1 and 0.2, respectively. Figure 3 shows the similar characteristic of the Indian pines image.

### 2.2.3. Removing Some Imaginary Parts of the Two Dimensional Quaternion Valued Singular Spectrum Analysis Components in the Last Group of the Color Planes

Since the zero color planes are inserted to generate the last quaternion valued matrix, the total number of the color planes is increased. To address this issue, the imaginary parts starting from the $k$ imaginary part of the two dimensional quaternion valued singular spectrum analysis components of the last group of the color planes are discarded. In particular, the total number of the imaginary parts of the two dimensional quaternion valued singular spectrum analysis components to be discarded is equal to the total number of inserted zero color planes.

### 2.3. Datasets

The first image is downloaded from the website of the Kennedy Space Center Florida (http://aviris.jpl.nasa.gov/). This image is called the KSC. It was acquired by the airborne visible/infrared imaging spectrometer in the National Aeronautics and Space Administration (NASA) on 23rd March 1996. In particular, the spectrometer is located at the position 20 km height from the ground and the spatial resolution is 18 m. The image consists of 224 frequency bands with the wavelengths being between 400 and 2500 nm. Additionally, the differences between two consecutive wavelengths are 10 nm. However, the frequency bands corresponding to the water absorption and those with the low signal to noise ratios are discarded. Eventually, there are only 176 frequency bands for performing the object recognition. Each color plane consists of $512 \times 614$ pixels. There are 13 different classes of objects representing various land types in the site environment.

The second image is downloaded from the website of the Indian Pines (http://aviris.jpl.nasa.gov/). It is the earliest test image used for performing the object recognition. Therefore, the Indian Pines is the second image used for evaluating the effectiveness of applying our proposed method for performing the object recognition. In 1992, the airborne visible infrared imaging spectrometer took an image of an Indian pine tree in Indiana US. Here, the size of the image is $145 \times 145$. It consists of 220 consecutive bands with the wavelengths of the frequency bands between 0.4 and 2.5 μm. However, because the 104th frequency band to the 108th frequency band, the 150th only employs the rest 200 frequency bands for performing the object recognition. There are 16 labeled land covered frequency bands to the 163th frequency band and the 220th frequency band cannot be reflected by water, which are mostly related to the agriculture, the forest and the perennial vegetation. However, only twelve classes are used for performing the object recognition because the total numbers of samples in some classes are too small.

The third image is acquired between 2001 and 2004 by the NASA EO-1 satellite placed at the Okavango Delta Botswana (http://www.ehu.eus/ccwintco/index.php?title=Hyperspectral_Remote_Sensing_Scenes). The Hyperion sensor on the NASA EO-1 satellite acquires the image with the spatial resolution 30 m over a 7.7 km strip. Each color plane consists of $1476 \times 256$ pixels. The image consists of 242 frequency bands with the

wavelengths of the frequency bands between 400 and 2500 nm. Likewise, the differences between two consecutive wavelengths are 10 nm. Some frequency bands are discarded. Eventually, only 145 frequency bands are employed for performing the object recognition. There are 14 different classes of objects representing various land types such as the seasonal swamps, the occasional swamps and the drier woodlands located in the distal portion of the delta.

## 3. Computer Numerical Simulation Results

### 3.1. Other Preprocessing Methods

First of all, the original hyperspectral images without any preprocessing were taken as the baselines for performing the comparison. In particular, the results obtained by the baseline approach were compared to those obtained by our proposed two dimensional quaternion valued singular spectrum analysis based preprocessing method and the other preprocessing methods.

Besides, since the two dimensional real valued singular spectrum analysis based method is similar to the two dimensional quaternion valued singular spectrum analysis based method, the two dimensional quaternion valued singular spectrum analysis based method was compared to the two dimensional real valued singular spectrum analysis based method. Moreover, since the hyperspectral images are the three dimensional signals, our proposed method was also compared to the three dimensional singular spectrum analysis based method [31,32]. Here, a three dimensional window moves along the hyperspectral image in the three dimensional manner. The detailed procedures for performing the three dimensional singular spectrum analysis are as follows. First, unfold each three dimensional portion of the hyperspectral image under the three dimensional window into a vector at each position where the window is located at. By moving the three dimensional window along the hyperspectral image, the column vectors are put both horizontally and vertically into a matrix. Eventually, a trajectory matrix is obtained. Then, apply the singular value decomposition to the trajectory matrix to obtain the three dimensional singular spectrum analysis components. After grouping the three dimensional singular spectrum analysis components and selecting these three dimensional singular spectrum analysis components, the hyperspectral image is reconstructed. Besides, there is another singular spectrum analysis called the multivariate two dimensional singular spectrum analysis. It is also used for processing the three dimensional signals. However, its operation likes that of the conventional two dimensional real valued singular spectrum analysis operating on each color plane. That is, the two dimensional window is employed. The sub-blocks in each color plane are unfolded into the column vectors. By moving the two dimensional window both horizontally and vertically, these column vectors are put both horizontally and vertically to form a matrix. By combining different matrices generated from different color plane, the trajectory matrix is obtained. In this paper, the multivariate two dimensional singular spectrum analysis based method is also compared to the two dimensional quaternion valued singular spectrum analysis based method. It is worth noting that the selection of the window size will affect the classification accuracy. In order to have a fair comparison, the window size of each singular spectrum analysis based method including the two dimensional real valued singular spectrum analysis based method, the multivariate two dimensional singular spectrum analysis based method, the three dimensional singular spectrum analysis based method and the two dimensional quaternion valued singular spectrum analysis based method is chosen in such a way that the highest classification accuracy is achieved in the corresponding method. Besides, the method for selecting the singular spectrum analysis components for each singular spectrum analysis based method is the same. That is, the singular spectrum analysis components are selected in such a way that the ratio of the interclass separation to the intraclass separation of the pixel vectors is maximized. Then, the selected singular spectrum analysis components are summed up together to construct the hyperspectral image. Finally, the pixel vectors of the reconstructed hyperspectral image are used for performing the object recognition.

In addition, since the principal component analysis based method is the most common method for performing the dimension reduction used for removing the redundancy in the hyperspectral image, our proposed method was also compared to the principal component analysis based method. Here, the dimensions of the pixel vectors in the hyperspectral image were reduced by 20%, 40% and 80%. Then, the pixel vectors with the reduced sizes were used for performing the object recognition. The obtained results were used for performing the comparison.

On the other hand, the median filtering based method is the most common color plane based method for performing the object recognition of the hyperspectral image. Therefore, our proposed method was also compared to the median filtering based method. Here, the median filter with different window sizes $3 \times 3$, $5 \times 5$, $10 \times 10$ and $20 \times 20$ was applied to all the color planes of the hyperspectral image. Then, the filtered pixel vectors were used for performing the object recognition. The obtained results were used for performing the comparison.

More importantly, since our proposed method is a kind of joint pixel domain and the spectral domain based method, the tensor decomposition based method was also compared. There are many different types of tensor decompositions in which the Tucker decomposition is the most common tensor decomposition. It is also known as the high order principal component analysis or the three dimensional principal component analysis. The Tucker decomposition is used to decompose the whole tensor into the weighted sum of the vector outer product of three vectors in three different directions as follows:

$$H \approx g \times_1 A \times_2 B \times_3 C = \sum_{p=1}^{P} \sum_{q=1}^{Q} \sum_{r=1}^{R} g_{pqr} a_p \circ b_q \circ c_r \tag{11}$$

Here, a hyperspectral image $H \in R^{I \times J \times K}$ is represented as a tensor. In particular, $A \in R^{I \times P}$, $B \in R^{J \times Q}$ and $C \in R^{K \times R}$ are the factor matrices, which are usually orthogonal to each other and they can be thought as the principal components of the tensor. The symbol "$\circ$" represents the operation of the vector outer product. The core tensor $g \in R^{P \times Q \times R}$ contains the weights of the vector outer product of these three vectors. For enhancing the hyperspectral image, the reconstructed hyperspectral image is a part of the weighted sum of the vector outer product of these three vectors as follows:

$$\widetilde{H} = \sum_{p=1}^{\vec{P}} \sum_{q=1}^{\vec{Q}} \sum_{r=1}^{\vec{R}} g_{pqr} a_p \circ b_q \circ c_r \tag{12}$$

Here, $0 \leq \vec{P} \leq P$, $0 \leq \vec{Q} \leq Q$ and $0 \leq \vec{R} \leq R$. In fact, the classification accuracy can be improved by the proper selection of the vector outer products of these three vectors. In this paper, the reconstructed hyperspectral image $\widetilde{H}$ was obtained by setting $\vec{P}$, $\vec{Q}$ and $\vec{R}$ to 10%, 20%, 40% and 80% of $P$, $Q$ and $R$, respectively. Then, the pixel vectors in $\widetilde{H}$ were used for performing the object recognition. Since the Tucker decomposition based method involves the weighted sum of the vector outer products, the required computational power is very large. Therefore, it is meaningful to compare the required computational power of our proposed method to that of the tensor based method for these three hyperspectral images. Here, the codes for implementing both our proposed method and the tensor based method were based on the Matlab version R2018b and executed in the computer with the Intel(R) Core(TM) i7-10700 CPU @ 2.90 GHz. In order to have a fair comparison, the total required processing time was evaluated only using one component for both our proposed method and the tensor based method. Table 2 shows the total required processing time of both our proposed method and the tensor based method for these three hyperspectral images. It is obvious to see from Table 2 that our proposed method requires the shorter processing times for all these three images compared to the Tucker decomposition based method.

**Table 2.** The comparison on the required processing times based on the quaternion valued singular spectrum analysis based method to that based on the Tucker decomposition based method.

| Methods | The Required Processing Times for Different Datasets | | |
|---|---|---|---|
| | KSC | Indian | Botswana |
| Our proposed method | 408.3331 s | 34.7727 s | 703.1345 s |
| Tucker decomposition based method | 1628.2310 s | 149.4909 s | 2835.1970 s |

Recently, the use of the CNN for performing the classification of the hyperspectral image becomes more and more popular. Therefore, our proposed method is also compared to the CNN based method. Since the hybrid SN [13] can exploit the information in both the pixel domain and the spectral domain, the hybrid SN based method is compared. In order to have a more fair comparison, the network parameters used in [13] were employed in the CNN for performing the comparison. That is, the lengths of the pixel vectors in the hyperspectral image are reduced by 15% and the epochs of the training model are set to 100.

*3.2. Classification of the Hyperspectral Image*

Performing the classification is to characterize or attribute the objects into the existing categories. The commonly used classification algorithms include the decision tree based method, the native Bayesian based method, the support vector machine based method, the neural network based method and the K nearest neighbor based method. For performing the object recognition of the hyperspectral image, the support vector machine based method is the most widely used classification based method because it is found to yield the best classification results compared to other classification based methods. Besides, there are several available libraries for implementing the support vector machine based method such as the LIBSVM toolbox in the Matlab [33]. Hence, this paper employed the support vector machine based method for performing the classification. Moreover, since the support vector machine with the Gaussian kernel usually achieves a good classification performance for the hyperspectral image, the support vector machine with the Gaussian kernel was used in our computer numerical simulations. Here, the penalty parameter and the gamma parameter were tuned every time through a grid search procedure. Then, the support vector machine was used for classifying the pixels vectors of the KSC image, the Indian pines image and the Botswana image into different classes.

*3.3. Performance Metrics*

In this paper, the overall classification accuracies under different conditions such as under different ratios of the total numbers of the pixel vectors in the training set to that in the overall image and under different method parameters were evaluated and compared. The results are shown in Tables 3–5. It is worth noting that the same method could yield different classification accuracies for different images under different conditions and different method parameters. Therefore, this paper employed a scoring system to rank the average performances of various methods for a given image at a given condition. The results are shown in Table 6. Here, the scores were ranked in the descending order. That is, the method with the highest score refers to the method achieving the highest classification accuracy. Finally, the overall score was computed for each method and this average score was employed as the metric for evaluating the average performance of the method. Furthermore, the macro F1 coefficient was also used as a performance metric. The results are shown in Table 7. The values of the macro F1 coefficients were between 0 and 1. Here, the larger of the value of the macro F1 coefficient refers to the training model being more fit for the data [34].

**Table 3.** The classification accuracy based on the KSC image.

| Different Preprocessing Methods with Different Parameters | | The Overall Classification Accuracies of Different Methods at Different Ratios of the Total Numbers of the Pixel Vectors in the Training Set to That in the Overall Image | | |
|---|---|---|---|---|
| | | Ratio of the Total Number of the Pixel Vectors in the Training Set to That in the Overall Image = 0.05 | Ratio of the Total Number of the Pixel Vectors in the Training Set to that in the Overall Image = 0.1 | Ratio of the Total Number of the Pixel Vectors in the Training Set to That in the Overall Image = 0.2 |
| Two dimensional quaternion valued singular spectrum analysis based method | The two dimensional quaternion valued singular spectrum analysis components are selected such that the ratio of the interclass separation to the intraclass separation of the pixel vectors is maximized. | 90.9604 (Score = 8) | 94.8466 (Score = 8) | 98.2519 (Score = 8) |
| Two dimensional real valued singular spectrum analysis based method [9] | The two dimensional real valued singular spectrum analysis components are selected such that the ratio of the interclass separation to the intraclass separation of the pixel vectors is maximized. | 86.2793 (Score = 5) | 90.0341 (Score = 5) | 93.8937 (Score = 4) |
| Multivariate two dimensional singular spectrum analysis based method [32] | The multivariate two dimensional singular spectrum analysis components are selected such that the ratio of the interclass separation to the intraclass separation of the pixel vectors is maximized. — All color planes of the hyperspectral image are processed together. | 57.0420 | 59.0716 | 65.4454 |
| | Four color planes of the hyperspectral image form a group and the color planes in each group are processed individually. | 89.5884 (Score = 7) | 90.5273 (Score = 6) | 96.2404 (Score = 5) |
| Three dimensional singular spectrum analysis based method [31,32] | The three dimensional singular spectrum analysis components are selected such that the ratio of the interclass separation to the intraclass separation of the pixel vectors is maximized. — All color planes of the hyperspectral image are processed together. | 89.3261 (Score = 6) | 92.3765 (Score = 7) | 97.2941 (Score = 6) |
| | Four color planes of the hyperspectral image form a group and the color planes in each group are processed individually. | 87.3083 | 90.6337 | 96.1207 |
| Principal component analysis based method [7] | The lengths of the pixel vectors are reduced by — 20% | 70.1977 | 72.8066 | 74.1858 |
| | 40% | 75.2825 | 78.5562 | 80.9387 |
| | 80% | 79.7417 (Score = 3) | 82.0486 (Score = 3) | 85.1054 (Score = 2) |
| Median filtering based method [4,9] | The window sizes are — 3 × 3 | 92.194 | 94.5132 | 97.8259 |
| | 5 × 5 | 93.3188 | 93.0229 | 97.3141 |
| | 10 × 10 | 93.5835 (Score = 9) | 95.0383 (Score = 9) | 97.9873 (Score = 7) |
| | 20 × 20 | 90.7869 | 92.3118 | 97.1185 |

Table 3. *Cont.*

| Different Preprocessing Methods with Different Parameters | | | The Overall Classification Accuracies of Different Methods at Different Ratios of the Total Numbers of the Pixel Vectors in the Training Set to That in the Overall Image | | |
| --- | --- | --- | --- | --- | --- |
| | | | Ratio of the Total Number of the Pixel Vectors in the Training Set to That in the Overall Image = 0.05 | Ratio of the Total Number of the Pixel Vectors in the Training Set to that in the Overall Image = 0.1 | Ratio of the Total Number of the Pixel Vectors in the Training Set to That in the Overall Image = 0.2 |
| Tucker decomposition based method [35] | The percentages of the P, Q and R are | 10% | 50.1211 (Score = 1) | 62.0954 (Score = 1) | 68.6782 (Score = 1) |
| | | 20% | 44.8951 | 53.0451 | 61.5182 |
| | | 40% | 37.6312 | 43.2069 | 48.3238 |
| | | 80% | 45.3793 | 51.2138 | 59.2912 |
| hybrid SN [13] | | | 69.6136 (Score = 2) | 79.8933 (Score = 2) | 98.3689 (Score = 9) |
| Baseline approach (without any preprocessing) | | | 84.6651 (Score = 4) | 89.5230 (Score = 4) | 92.2653 (Score = 3) |

Table 4. The classification accuracy based on the Indian image.

| Different Preprocessing Methods with Different Parameters | | The Overall Classification Accuracies of Different Methods at Different Ratios of the Total Numbers of the Pixel Vectors in the Training Set to That in the Overall Image | | |
| --- | --- | --- | --- | --- |
| | | Ratio of the Total Number of the Pixel Vectors in the Training Set to That in the Overall Image = 0.05 | Ratio of the Total Number of the Pixel Vectors in the Training Set to That in the Overall Image = 0.1 | Ratio of the Total Number of the Pixel Vectors in the Training Set to That in the Overall Image = 0.2 |
| Two dimensional quaternion valued singular spectrum analysis based method | The two dimensional quaternion valued singular spectrum analysis components are selected such that the ratio of the interclass separation to the intraclass separation of the pixel vectors is maximized. | 94.6157 (Score = 9) | 97.3960 (Score = 8) | 98.3371 (Score = 7) |
| Two dimensional real valued singular spectrum analysis based method [9] | The two dimensional real valued singular spectrum analysis components are selected such that the ratio of the interclass separation to the intraclass separation of the pixel vectors is maximized. | 90.8834 (Score = 5) | 95.2775 (Score = 6) | 98.4984 (Score = 9) |

**Table 4.** *Cont.*

| Different Preprocessing Methods with Different Parameters | | | The Overall Classification Accuracies of Different Methods at Different Ratios of the Total Numbers of the Pixel Vectors in the Training Set to That in the Overall Image | | |
|---|---|---|---|---|---|
| | | | Ratio of the Total Number of the Pixel Vectors in the Training Set to That in the Overall Image = 0.05 | Ratio of the Total Number of the Pixel Vectors in the Training Set to That in the Overall Image = 0.1 | Ratio of the Total Number of the Pixel Vectors in the Training Set to That in the Overall Image = 0.2 |
| Multivariate two dimensional singular spectrum analysis based method [32] | The multivariate two dimensional singular spectrum analysis components are selected such that the ratio of the interclass separation to the intraclass separation of the pixel vectors is maximized. | All color planes of the hyperspectral image are processed together. | 49.9111 | 50.8882 | 51.0549 |
| | | Four color planes of the hyperspectral image form a group and the color planes in each group are processed individually. | 91.0612 (Score = 6) | 95.8402 (Score = 7) | 97.5552 (Score = 5) |
| Three dimensional singular spectrum analysis based method [31,32] | The three dimensional singular spectrum analysis components are selected such that the ratio of the interclass separation to the intraclass separation of the pixel vectors is maximized. | All color planes of the hyperspectral image are processed together. | 91.8453 (Score = 7) | 94.3617 (Score = 5) | 97.6297 (Score = 6) |
| | | Four color planes of the hyperspectral image form a group and the color planes in each group are processed individually. | 91.5365 | 93.2210 | 97.2084 |
| Principal component analysis based method [7] | The lengths of the pixel vectors are reduced by | 20% | 73.4135 | 78.2522 | 81.1740 |
| | | 40% | 73.9676 | 79.7859 (Score = 1) | 83.4202 |
| | | 80% | 75.4522 (Score = 2) | 79.5763 | 85.1328 (Score = 1) |
| Median filtering based method [4,9] | The window sizes are | 3 × 3 | 812546 | 88.3041 | 90.9531 |
| | | 5 × 5 | 85.2274 | 89.6502 | 93.5468 |
| | | 10 × 10 | 87.7888 (Score = 4) | 92.5742 (Score = 4) | 93.1745 |
| | | 20 × 20 | 86.5029 | 90.4667 | 94.1549 (Score = 4) |
| Tucker decomposition based method [35] | The percentages of the P, Q and R are | 10% | 82.7182 (Score = 3) | 88.3372 (Score = 3) | 92.5540 |
| | | 20% | 81.3696 | 87.2228 | 92.7277 (Score = 3) |
| | | 40% | 78.2122 | 85.1263 | 87.6768 |
| | | 80% | 75.9435 | 81.4190 | 84.6860 |

**Table 4.** *Cont.*

| Different Preprocessing Methods with Different Parameters | | The Overall Classification Accuracies of Different Methods at Different Ratios of the Total Numbers of the Pixel Vectors in the Training Set to That in the Overall Image | | |
|---|---|---|---|---|
| | | Ratio of the Total Number of the Pixel Vectors in the Training Set to That in the Overall Image = 0.05 | Ratio of the Total Number of the Pixel Vectors in the Training Set to That in the Overall Image = 0.1 | Ratio of the Total Number of the Pixel Vectors in the Training Set to That in the Overall Image = 0.2 |
| hybrid SN [13] | | 93.5503 (Score = 8) | 98.1896 (Score = 9) | 98.3414 (Score = 8) |
| Baseline approach (without any preprocessing) | | 75.0742 (Score = 1) | 81.3721 (Score = 2) | 85.6658 (Score = 2) |

**Table 5.** The classification accuracy based on the Botswana image.

| Different Preprocessing Methods with Different Parameters | | | The overall Classification Accuracies of Different Methods at Different Ratios of the Total Numbers of the Pixel Vectors in the Training Set to That in the Overall Image | | |
|---|---|---|---|---|---|
| | | | Ratio of the Total Number of the Pixel Vectors in the Training Set to that in the Overall Image = 0.05 | Ratio of the Total Number of the Pixel Vectors in the Training Set to That in the Overall Image = 0.1 | Ratio of the Total Number of the Pixel Vectors in the Training Set to that in the Overall Image = 0.2 |
| Two dimensional quaternion valued singular spectrum analysis based method | The two dimensional quaternion valued singular spectrum analysis components are selected such that the ratio of the interclass separation to the intraclass separation of the pixel vectors is maximized. | | 97.5097 (Score = 9) | 99.0106 (Score = 9) | 99.6933 (Score = 9) |
| Two dimensional real valued singular spectrum analysis based method [9] | The two dimensional real valued singular spectrum analysis components are selected such that the ratio of the interclass separation to the intraclass separation of the pixel vectors is maximized. | | 96.2807 (Score = 7) | 98.0894 (Score = 5) | 98.4279 (Score = 5) |
| Multivariate two dimensional singular spectrum analysis based method [32] | The multivariate two dimensional singular spectrum analysis components are selected such that the ratio of the interclass separation to the intraclass separation of the pixel vectors is maximized. | All color planes of the hyperspectral image are processed together. | 53.2018 | 58.8195 | 62.1933 |
| | | Four color planes of the hyperspectral image form a group and the color planes in each group are processed individually. | 96.8305 (Score = 8) | 98.2259 (Score = 6) | 99.2331 (Score = 8) |

Table 5. *Cont.*

| Different Preprocessing Methods with Different Parameters | | | The overall Classification Accuracies of Different Methods at Different Ratios of the Total Numbers of the Pixel Vectors in the Training Set to That in the Overall Image | | |
|---|---|---|---|---|---|
| | | | Ratio of the Total Number of the Pixel Vectors in the Training Set to that in the Overall Image = 0.05 | Ratio of the Total Number of the Pixel Vectors in the Training Set to That in the Overall Image = 0.1 | Ratio of the Total Number of the Pixel Vectors in the Training Set to that in the Overall Image = 0.2 |
| Three dimensional singular spectrum analysis based method [31,32] | The three dimensional singular spectrum analysis components are selected such that the ratio of the interclass separation to the intraclass separation of the pixel vectors is maximized. | All color planes of the hyperspectral image are processed together. | 94.5019 (Score = 5) | 98.2600 (Score = 7) | 99.0798 (Score = 6) |
| | | Four color planes of the hyperspectral image form a group and the color planes in each group are processed individually. | 93.0246 | 98.3654 | 98.9647 |
| Principal component analysis based method [7] | The lengths of the pixel vectors are reduced by | 20% | 88.5511 | 91.7434 | 94.2868 |
| | | 40% | 88.8745 | 92.6987 (Score = 2) | 94.4018 |
| | | 80% | 90.2975 (Score = 2) | 92.5623 | 94.7853 (Score = 2) |
| Median filtering based method [4,9] | The window sizes are | 3 × 3 | 92.0440 | 95.8376 | 97.8528 (Score = 3) |
| | | 5 × 5 | 96.0220 | 97.6117 | 96.1123 |
| | | 10 × 10 | 96.2807 (Score = 6) | 98.3623 (Score = 8) | 94.3540 |
| | | 20 × 20 | 93.7904 | 95.7352 | 95.6848 |
| Tucker decomposition based method [35] | The percentages of the P, Q and R are | 10% | 89.9094 | 95.3599 | 95.4371 |
| | | 20% | 90.3946 | 95.5647 (Score = 3) | 98.0445 (Score = 4) |
| | | 40% | 94.1462 (Score = 4) | 95.1552 | 97.3160 |
| | | 80% | 89.7801 | 93.0740 | 96.1273 |
| hybrid SN [13] | | | 92.5145 (Score = 3) | 97.9480 (Score = 4) | 99.2304 (Score = 7) |
| Baseline approach (without any preprocessing) | | | 85.9314 (Score = 1) | 91.8458 (Score = 1) | 93.4049 (Score = 1) |

**Table 6.** The average scores of different methods for different images.

| Methods | The Total Scores of Different Methods for Different Images | | | The Average Total Scores |
|---|---|---|---|---|
| | KSC | Indian | Botswana | |
| Two dimensional quaternion valued singular spectrum analysis based method | 24 | 24 | 27 | 25 |
| Two dimensional real valued singular spectrum analysis based method [9] | 14 | 20 | 17 | 17 |
| Multivariate two dimensional singular spectrum analysis based method [32] | 18 | 18 | 22 | 19.3333 |
| Three dimensional singular spectrum analysis based method [31,32] | 19 | 18 | 18 | 18.3333 |
| Principal component analysis based method [7] | 8 | 4 | 6 | 6 |
| Median filtering based method [4,9] | 25 | 12 | 17 | 18 |
| Tucker decomposition based method [35] | 3 | 9 | 11 | 7.6667 |
| hybrid SN [13] | 13 | 25 | 14 | 17.3333 |
| Baseline approach (without any preprocessing) | 11 | 5 | 3 | 6.3333 |

**Table 7.** The macro F1 coefficients of different methods for different images.

| Methods | Macro F1 Coefficients of Different Methods at Different Ratios of the Total Numbers of the Pixel Vectors in the Training Set to that in the Overall Images | | | | | | | | |
|---|---|---|---|---|---|---|---|---|---|
| | KSC | | | Indian | | | Botswana | | |
| | 0.05 | 0.1 | 0.2 | 0.05 | 0.1 | 0.2 | 0.05 | 0.1 | 0.2 |
| Two dimensional quaternion valued singular spectrum analysis based method | 0.8719 | 0.9322 | 0.9773 | 0.9493 | 0.9759 | 0.9857 | 0.9731 | 0.9909 | 0.9971 |
| Two dimensional real valued singular spectrum analysis based method [9] | 0.8217 | 0.8634 | 0.9248 | 0.9185 | 0.9599 | 0.9878 | 0.9577 | 0.9765 | 0.9827 |
| Multivariate two dimensional singular spectrum analysis based method [32] | 0.8251 | 0.9118 | 0.9327 | 0.9065 | 0.9621 | 0.9745 | 0.9637 | 0.9770 | 0.9903 |
| Three dimensional singular spectrum analysis based method [31,32] | 0.8282 | 0.8789 | 0.951 | 0.9194 | 0.9446 | 0.9793 | 0.9177 | 0.9833 | 0.9906 |
| Principal component analysis based method [7] | 0.6972 | 0.7393 | 0.7906 | 0.7256 | 0.7923 | 0.8439 | 0.9095 | 0.93 | 0.8439 |
| Median filtering based method [4,9] | 0.8974 | 0.9291 | 0.9562 | 0.8974 | 0.9245 | 0.9424 | 0.9581 | 0.9824 | 0.9803 |
| Tucker decomposition based method [35] | 0.4536 | 0.5748 | 0.6566 | 0.8207 | 0.897 | 0.9366 | 0.9468 | 0.959 | 0.9803 |
| hybrid SN [13] | 0.6321 | 0.8035 | 0.9811 | 0.9451 | 0.9810 | 0.9821 | 0.9231 | 0.9754 | 0.9905 |

## 4. Discussions

For the KSC image, the Indian pines image and the Botswana image, the classification accuracies under different conditions and different method parameters are given in Tables 3–5,

respectively. Table 6 shows the average scores of various methods. Here, the parameters used in various methods are also shown in the tables.

Compared to the classification accuracies obtained by the two dimensional quaternion valued singular spectrum analysis based method to the baseline approach without any preprocessing, it can be seen from Table 3 that the classification accuracies for the KSC image increased from 84.6651% to 90.9604%, from 89.5230% to 94.8466% and from 92.2653% to 98.2519% for the ratios of the total numbers of the pixel vectors in the training set to that in the overall image being equal to 0.05, 0.1 and 0.2, respectively. Likewise, it can be seen from Table 4 that the classification accuracies for the Indian pines image increased from 75.0742% to 94.6157%, from 81.3721% to 97.3960% and from 85.6658% to 98.3371% for the ratios of the total numbers of the pixel vectors in the training set to that in the overall image being equal to 0.05, 0.1 and 0.2, respectively. Similar, it can be seen from Table 5 that the classification accuracies for the Botswana image increase from 85.9314% to 97.5097%, from 91.8458% to 99.0106% and from 93.4049% to 99.6933% for the ratios of the total numbers of the pixel vectors in the training set to that in the overall image being equal to 0.05, 0.1 and 0.2, respectively. It can be concluded that the two dimensional quaternion valued singular spectrum analysis based method consistently achieves the higher classification accuracies compared to the baseline approach. This is because the two dimensional quaternion valued singular spectrum analysis based method exploits the correlations among the pixels in each color plane and that among various color planes. As both the spatial information and the spectral information of the image are exploited, better results can be obtained. Moreover, the two dimensional quaternion valued singular spectrum analysis components are selected in such a way that the ratio of the interclass separation to the intraclass separation is maximized. This also accounts for obtaining the better results. On the other hand, the baseline approach does not perform any preprocessing. Therefore, the baseline approach consistently achieves the lower classification accuracies compared to the two dimensional quaternion valued singular spectrum analysis based method.

Compared to the classification accuracies obtained by the two dimensional quaternion valued singular spectrum analysis based method to the two dimensional real valued singular spectrum analysis based method, it can be seen from Table 3 that the two dimensional quaternion valued singular spectrum analysis based method achieved the higher classification accuracies for the KSC image by 5.4255%, 5.3452% and 4.6416% at the ratios of the total numbers of the pixel vectors in the training set to that in the overall image being equal to 0.05, 0.1 and 0.2, respectively. For the Indian pines image and the Botswana image, the two dimensional quaternion valued singular spectrum analysis based method achieved the higher classification accuracies compared to the two dimensional real valued singular spectrum analysis based method by the average of 2.0555% and 1.1671%, respectively. It can be concluded that the two dimensional quaternion valued singular spectrum analysis based method overall achieved the higher classification accuracies compared to the two dimensional real valued singular spectrum analysis based method. This is because the two dimensional real valued singular spectrum analysis based method only exploited the spatial information within a color plane, while the two dimensional quaternion valued singular spectrum analysis based method exploited both the spatial information within a color plane and the spectral information among various color planes. Therefore, the two dimensional quaternion valued singular spectrum analysis based method yielded the higher classification accuracies compared to the two dimensional real valued singular spectrum analysis based method. Although the two dimensional real valued singular spectrum analysis based method yielded the higher classification accuracy compared to the two dimensional quaternion valued singular spectrum analysis based method when the ratio of the total number of the pixel vectors in the training set to that in the overall image being equal to 0.2 for the Indian pines image, this improvement was so mild that it can be ignored. On the other hand, the two dimensional real valued singular spectrum analysis based method yielded the higher classification accuracies compared to the baseline approach.

For the results obtained by the multivariate two dimensional singular spectrum analysis based method, it can be seen that the baseline approach even outperformed the multivariate two dimensional singular spectrum analysis based method when the multivariate two dimensional singular spectrum analysis was applied to all the color planes. This is because that the multivariate two dimensional singular spectrum analysis processes the hyperspectral image slices by slices to construct the trajectory matrix. In this case, the structure of the three dimensional hyperspectral image was ignored or even destroyed. This yields the bad results. It is worth noting that the classification accuracy improved when the multivariate two dimensional singular spectrum analysis was only applied to four color planes each time. However, our proposed method still achieved the higher classification accuracies compared to the multivariate two dimensional singular spectrum analysis based method operating only on four color planes each time by the average of 2.7976%, 2.1094% and 0.6547% for the KSC image, the Indian pines image and the Botswana image, respectively. This is because the quaternion algebraic could deeply exploit the relationship among the pixels within a color plane and those among different color planes of the hyperspectral image.

Compared the classification accuracies obtained by the two dimensional quaternion valued singular spectrum analysis based method to the three dimensional singular spectrum analysis based method, it can be seen from Table 3 that the classification accuracies for the KSC image increased from 89.3261% to 90.9604%, from 92.3765% to 94.8466% and from 97.2941% to 98.2519% for the ratios of the total numbers of the pixel vectors in the training set to that in the overall image being equal to 0.05, 0.1 and 0.2, respectively. Likewise, it can be seen from Table 4 that the classification accuracies for the Indian pines image increased from 91.8453% to 94.6157%, from 94.3617% to 97.3960% and from 97.6297% to 98.3371% for the ratios of the total numbers of the pixel vectors in the training set to that in the overall image being equal to 0.05, 0.1 and 0.2, respectively. Similarly, it can be seen from Table 5 that the classification accuracies for the Botswana image increase from 94.5019% to 97.5097%, from 98.2600% to 99.0106% and from 99.0798% to 99.6933% for the ratios of the total numbers of the pixel vectors in the training set to that in the overall image being equal to 0.05, 0.1 and 0.2, respectively. It can be concluded that the two dimensional quaternion valued singular spectrum analysis based method consistently achieved the higher classification accuracies compared to the three dimensional singular spectrum analysis based method.

Compared to the classification accuracies obtained by the two dimensional quaternion valued singular spectrum analysis based method to the principal component analysis based method, the two dimensional quaternion valued singular spectrum analysis based method achieved the higher classification accuracies by the average of 15.0381%, 20.9934% and 6.6581% for the KSC image, the Indian pines image and the Botswana image, respectively. Here, the reduction ratio on the length of the pixel vectors in the principal component analysis based method was chosen in such a way that the highest classification accuracy was achieved for a given image. Besides, it is found that the principal component analysis based method yielded the lower classification accuracies compared to the other preprocessing methods. This is because the principal component analysis based method only exploits the spectral information among various color planes, but the correlation among the pixel vectors within the same color plane has not been exploited. As a result, the principal component analysis based method yielded the lower classification accuracies compared to other methods.

Compared to the classification accuracies obtained by the two dimensional quaternion valued singular spectrum analysis based method to the Tucker decomposition based method, the two dimensional quaternion valued singular spectrum analysis based method achieved the higher classification accuracies by the average of by 59.0953%, 10.2291% and 2.9534% for the KSC image, the Indian pines image and the Botswana image, respectively. Here, the percentages of the P, Q and R in the Tucker decomposition based method were chosen in such a way that the highest classification accuracy was achieved for a given

image. It is worth noting that the Tucker decomposition based method even yielded a lower classification accuracy compared to the baseline approach for the KSC image.

Compared to the classification accuracies obtained by the two dimensional quaternion valued singular spectrum analysis based method to the median filtering based method, the two dimensional quaternion valued singular spectrum analysis based method achieved a lower classification accuracy by the average of 0.9115% for the KSC image. However, the two dimensional quaternion valued singular spectrum analysis based method achieved the higher classification accuracies by the average of 5.8090% and 1.2722% for the Indian pines image and the Botswana image, respectively. Here, the window sizes of the median filtering based method were chosen in such a way that the highest classification accuracy was achieved for a given image. It can be seen that the median filtering based method was not robust. This is because the median filtering based method did not achieve the higher classification accuracies compared to our proposed method for all the images. Although the median filtering based method yielded the highest classification accuracy for the KSC image at the ratios of the total numbers of the pixel vectors in the training set to that in the overall image being equal to 0.05 and 0.1, the classification accuracy at the ratio of the total number of the pixel vectors in the training set to that in the overall image being equal to 0.2 was lower than that based on the two dimensional quaternion valued singular spectrum analysis based method. It is worth noting that the median filtering based method is a spatial domain based nonlinear denoising technique exploiting the statistic of the pixel values within a color plane. It replaces a pixel value by the median value of its neighborhood pixel values. Although it can effectively suppress the impulsive noise, the edges in an image cannot be retained. This is because the median values of the pixel values around the edge points will not give the pixel values of the edges. However, as there are many objects in the hyperspectral image and each object contains many edges, the medium filtering based method in general yielded the lower classification accuracy.

From Tables 3–6, it can be concluded that the CNN based method (hybrid SN) performed well when the ratios of the total numbers of the pixel vectors in the training set to that in the overall image were high. However, the CNN based method performed poorly when the ratios of the total numbers of the pixel vectors in the training set to that in the overall image were low. It is even worse than the baseline approach for the KSC dataset. In fact, this is a common problem of the CNN based method. Compared to the classification accuracies obtained by the two dimensional quaternion valued singular spectrum analysis based method to the hybrid SN based method, the two dimensional quaternion valued singular spectrum analysis based method achieved the higher classification accuracies by the average of by 16.4208%, 0.1087% and 2.3169% for the KSC image, the Indian pines image and the Botswana image, respectively. The results show that our proposed method is more robust for the ratios of the total numbers of the pixel vectors in the training set to that in the overall image.

From Table 6, it can be seen that all the above preprocessing methods except the PCA based method could improve the classification accuracies. This is because the average scores of these preprocessing methods were higher than that of the baseline approach. Among all the preprocessing methods, the two dimensional quaternion valued singular spectrum analysis based method achieved the highest average score for two images out of three images. This implies that the two dimensional quaternion valued singular spectrum analysis based method did not only yield the higher classification accuracy compared to other preprocessing methods, but it was also robust for most of the images.

To compare the balance degree of the model trained by our proposed method to those trained by the other methods, Table 7 shows the macro F1 coefficients based on the best results obtained by each method under different conditions. It can be seen from Table 7 that the macro F1 coefficients of the models trained by our proposed method outperformed those trained by the other methods. This implies that the models trained by our proposed method were more balanced than those trained by the other methods.

## 5. Conclusions

This paper employed the two dimensional quaternion valued singular spectrum analysis based method for enhancing the hyperspectral image for performing the object recognition. In particular, the two dimensional quaternion valued singular spectrum analysis components are selected in such a way that the ratio of the interclass separation to the intraclass separation of the pixel vectors was maximized. Here, the correlation among the pixels within each color plane and that among various color planes were exploited. The computer numerical simulation results show that the two dimensional quaternion valued singular spectrum analysis based method achieved the higher classification accuracy compared to other preprocessing methods such as the two dimensional real valued singular spectrum analysis based method, the three dimensional singular spectrum analysis based method, the multivariate two dimensional singular spectrum analysis based method, the principal component analysis based method, the median filtering based method, the Tucker decomposition based method and the hybrid SN based method. Additionally, the two dimensional quaternion valued singular spectrum analysis based method was robust for most of the images and the ratios of the total numbers of the pixel vectors in the training set to that in the overall image.

**Author Contributions:** Conceptualization, Y.L.; methodology, Y.L.; software, Y.L.; validation, B.W.-K.L., L.H., Y.Z., N.X., X.Z. and X.W.; formal analysis, Y.L.; investigation, Y.L.; resources, Y.L.; data curation, Y.L.; writing—original draft preparation, Y.L.; writing—review and editing, B.W.-K.L.; visualization, Y.L.; supervision, B.W.-K.L.; project administration, B.W.-K.L.; funding acquisition, B.W.-K.L. All authors have read and agreed to the published version of the manuscript.

**Funding:** This paper was supported partly by the National Nature Science Foundation of China, grant numbers U1701266, 61671163 and 62071128, the Team Project of the Education Ministry of the Guangdong Province, grant number 2017KCXTD011, the Guangdong Higher Education Engineering Technology Research Center for Big Data on Manufacturing Knowledge Patent, grant number 501130144, and Hong Kong Innovation and Technology Commission, Enterprise Support Scheme S/E/070/17.

**Institutional Review Board Statement:** Not applicable.

**Informed Consent Statement:** Not applicable.

**Data Availability Statement:** The data is obtained from the following: KSC: http://aviris.jpl.nasa.gov/; Indian Pines: http://aviris.jpl.nasa.gov/; Botswana: http://www.ehu.eus/ccwintco/index.php?title=Hyperspectral_Remote_Sensing_Scenes.

**Conflicts of Interest:** There is no conflict of interest.

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
