# Peer review of "Hyperspectral Image Enhancement by Two Dimensional Quaternion Valued Singular Spectrum Analysis for Object Recognition"

_remotesensing, doi:10.3390/rs13030405_

Round 1
Reviewer 1 Report
Object recognition via enhancing hyperspectral image by two dimensional quaternion valued singular spectrum analysis
The authors have proposed a 2-D quaternion valued singular spectrum analysis method to enhance hyperspectral images for object recognition. Four color planes were used for image enhancement when compared to the orthodox spectrum analysis methods that have utilized only pixels in independent color planes. The contribution of the work is apparent in the abstract and the manuscript is easy to follow. However, I would like the authors to adequately address the following issues before the manuscript can be recommended for possible publication.
- Title of manuscript should be changed to “Hyperspectral image enhancement by two-dimensional quaternion valued singular spectrum analysis for object recognition” to better convey the intended message.
- Replace the definition of a quaternion by “a quaternion valued number is a number having three imaginary components with one real componet”. Only pure quaternions have no real component.
- Section titles are too long in most cases. Change title of Section 2 to “Review of 2D quaternion valued singular spectrum analysis”. Change title of Section 3 to “The proposed method”. Title of Section 4 should be changed to “Experimental results and Discussions”. Title of Section 4.2 should be changed to “Pre-processing method”. Title of Sections 4.4 and 4.5 should be changed to “Performance metrics” and “Results”, respectively.
- Since Section 2 is a review, it is expected that authors would sufficiently make references to literature. The section is currently devoid of references to justify claims. For instance, claims such as “it is worth noting that this approach has been applied to many science and engineering applications successfully” and “… the de-Hankelization operation over ….” certainly require citations. The entire Section 2 is missing important citations to strengthen argument.
- Sections 3 and 4 are ill-structured. What is the source of equation (10)? The statements starting from “Figure 1 to Figure 3 show …” are misplaced. Results should be presented in the result section, not in the method section and vice versa. This anomaly permeates Sections 3 and 4. Sections 4.2, 4.3, 4.4 should be moved to the method section and numbered appropriately.
- The authors should separate results from discussion. Section 4 may be subdivided as follows. 4.1 Experimental hyperspectral images, 4.2 Results, 4.2.1 Qualitative result, 4.2.2 Quantitative result. 4.3 Discussions. Since the authors are claiming their method enhances hyperspectral images, qualitative results become important. Authors should present visual results in comparison to other existing methods.
- The authors have mentioned in the last statement of third paragraph in Section 1 that the computational power of tensor-based decomposition method is very high without literature support. Hence, it becomes expedient for the authors to provide an algorithmic description of their method. Moreover, they should further compare the results of their method with those of tensor-based methods in terms of computational times.
- Why the authors have used only accuracy performance metric? Why not consider other benchmarked metrics such as specificity, sensitivity, recognition error and receiver operating characteristics curve?
- There are confusing statements in Section 4.1. It is confusing to say “Three hyperspectral images acquired…” I guess the authors are referring to three different categories of hyperspectral image databases or corpora (NASA, Indian Pines, Botswana). Each image data corpus should be described in clear terms in tandem. The authors should provide references or web links to the databases.
- The last paragraph in section 4.2 is not appropriate in the method section and should be deleted.
- The first paragraph in section 4.3 is not appropriate for the section theme and should be deleted.
- The symbols for penalty and gamma parameters should not be written because authors are assuming that readers are familiar with their implementation tool in section 4.3. Simple statement such as “the penalty and gamma parameters were tuned …” is sufficient. The authors can further explain the physical interpretation of the parameters, but it is not important to list their symbols.
- The authors should provide references to other pre-processing methods that were compared quantitatively in tables 1 to 4.
- The kind of artefacts in the hyperspectral images that were enhanced by the proposed method were not explained in the manuscript.
- Authors should define all acronyms where they were first used. They should not assume that readers are familiar with them, for instance NASA.
Author Response
Dear Reviewer1, Please see the attachment for the full paper.
Response to Reviewer 1 Comments
Point 1 : Title of manuscript should be changed to “Hyperspectral image enhancement by two-dimensional quaternion valued singular spectrum analysis for object recognition” to better convey the intended message.
Response 1: Agree.
The title has been changed to “Hyperspectral image enhancement by two dimensional quaternion valued singular spectrum analysis for object recognition”.
Point 2: Replace the definition of a quaternion by “a quaternion valued number is a number having three imaginary components with one real component”. Only pure quaternions have no real component.
Response 2: Agree.
The definition in the introduction has been changed.
Point 3: Section titles are too long in most cases. Change title of Section 2 to “Review of 2D quaternion valued singular spectrum analysis”. Change title of Section 3 to “The proposed method”. Title of Section 4 should be changed to “Experimental results and Discussions”. Title of Section 4.2 should be changed to “Pre-processing method”. Title of Sections 4.4 and 4.5 should be changed to “Performance metrics” and “Results”, respectively.
Response 3: Agree.
The title has been changed and the structure of the paper has been reorganized according to the reviewers’ comments and the publisher.
Point 4: Since Section 2 is a review, it is expected that authors would sufficiently make references to literature. The section is currently devoid of references to justify claims. For instance, claims such as “it is worth noting that this approach has been applied to many science and engineering applications successfully” and “… the de-Hankelization operation over ….” certainly require citations. The entire Section 2 is missing important citations to strengthen argument.
Response 4: Agree.
The citations to strengthen argument has been claimed in the last paragraph of the section 2.1.3 :
It is worth to noting that the embedding and the de-hankelization stages are both the core operations to all the singular spectrum analysis technique, and has been applied to many science and engineering applications successfully[26] [27]. Moreover, the SSA window shows the similar characteristic to the CNN window, which is used to extract the local relationship among the pixels in the window. Besides, the quaternion singular value decomposition, as the extension of the singular value decomposition, has been successfully applied to many multichannel application, such as processing of color image [28] [29].
Point 5: Sections 3 and 4 are ill-structured. What is the source of equation (10)? The statements starting from “Figure 1 to Figure 3 show …” are misplaced. Results should be presented in the result section, not in the method section and vice versa. This anomaly permeates Sections 3 and 4. Sections 4.2, 4.3, 4.4 should be moved to the method section and numbered appropriately.
Response 5: Agree.
First, the definition of the equation (10) has been clearly claimed in the third paragraph of 2.2.3:
Define the ratio of the interclass separation to the intraclass separation of the pixel vectors be:
, (10)
for be the mean of the vectors of class and be the vector of class.
The statement of Figure 1-3 has been claimed in 2.2.3. The structure and the titles of the paper has been reorganized according to the reviewer comments and the publisher.
Point 6: The authors should separate results from discussion. Section 4 may be subdivided as follows. 4.1 Experimental hyperspectral images, 4.2 Results, 4.2.1 Qualitative result, 4.2.2 Quantitative result. 4.3 Discussions. Since the authors are claiming their method enhances hyperspectral images, qualitative results become important. Authors should present visual results in comparison to other existing methods.
Response 6: Partially agree.
The structure and the titles of the paper has been reorganized according to the reviewer comments and the publisher.
We may misunderstand your comments. The term enhancement does neither for improving the visual quality nor suppressing the artifacts. Therefore, presenting the visual quality of the hyperspectral image is not fair for comparing the results. On the other hand, as the enhancement is for performing the object recognition, it is more fair to compare the results from the classification viewpoint.
Point 7: The authors have mentioned in the last statement of third paragraph in Section 1 that the computational power of tensor-based decomposition method is very high without literature support. Hence, it becomes expedient for the authors to provide an algorithmic description of their method. Moreover, they should further compare the results of their method with those of tensor-based methods in terms of computational times.
Response 7: Agree,.
The comparison results of the tensor based method and the proposed method have been put in the Table2 and the algorithmic description is put in the last 6-8 paragraphs of the section 3.1:
More importantly, since the proposed method is a kind of joint pixel domain and spectral domain based method, the tensor decomposition based method is also compared. There are many different types of tensor decompositions in which the Tucker decomposition is the most common tensor decomposition. It is also known as the high order principle component analysis or the three dimensional principle component analysis. The Tucker decomposition is used to decompose the whole tensor into the weighted sum of the vector outer product of three vectors in three different directions as follow:
. (11)
Here, a hyperspectral image is represented as a tensor. In particular, , and are the factor matrices which are usually orthogonal and can be thought of as the principal components of the tensor. The symbol ‘’ represents the operation of the vector outer product. The core tensor contains the weights of the vector outer product of these three vectors. For enhancing the hyperspectral image, the reconstructed hyperspectral image is the sum of a part of the weighted sum of the vector outer product of these three vectors as follows:
. (12)
Here, , and . In fact, the classification accuracy can be improved by the proper selection of the vector outer product components of these three vectors. In this paper, the reconstructed hyperspectral image is obtained by setting , and to 10%, 20%, 40% and 80% of , and , respectively. Then, the pixel vectors in are used for performing the object recognition. Since the tucker decomposition based method involves the multiplication among each modes , the cost of the computational power is very large, it is meaningful to compare the computation times for each hyperspectral image. Table 2 shows the total time cost by the proposed method and the tensor based method for the three hyperspectral images (For the fair comparison, the computation time is computed as only one components be selected for the two methods). The computer used for experiment use the Intel(R) Core(TM) i7-10700 CPU @ 2.90GHz, the Matlab of version R2018b.
Table 2. The comparison of computational time between the quaternion valued singular spectrum analysis and the tucker decomposition
|
Method |
The computational time in different data set |
||
|
KSC |
Indian |
Botswana |
|
|
The proposed method |
408.3331 seconds |
34.7727 seconds |
703.1345 seconds |
|
Tucker decomposition |
1628.2310 seconds |
149.4909 seconds |
2835.1970 seconds |
Point 8 : Why the authors have used only accuracy performance metric? Why not consider other benchmarked metrics such as specificity, sensitivity, recognition error and receiver operating characteristics curve?
Response 8: Agree.
Since the experiment we face is the multi class problem, we use the Maro-F1 as another performance metric, shown in Table 7 :
Furthermore, for the more convincing results, we consider the Maro-F1 coefficient for benchmarked metrics (Table 7), which is from 0 to 1. It means that the larger of the Maro-F1 coefficient is, the training model is more fit for the data. [35]: It can be seen from Table7 that the Macro F1 coefficient of the models trained by our proposed method outperforms most of the others, which means that the models trained by the proposed method are more balanced than the other methods.
|
Method |
Macro-F1 of different methods at different ratios of the total number of the pixel vectors in the training set to that in the different dataset |
||||||||
|
KSC |
Indian |
Botswana |
|||||||
|
0.05 |
0.1 |
0.2 |
0.05 |
0.1 |
0.2 |
0.05 |
0.1 |
0.2 |
|
|
Two dimensional quaternion valued singular spectrum analysis based method |
0.8719
|
0.9322
|
0.9773
|
0.9493
|
0.9759
|
0.9857
|
0.9731
|
0.9909
|
0.9971
|
|
Two dimensional real valued singular spectrum analysis based method [9] |
0.8217
|
0.8634
|
0.9248
|
0.9185
|
0.9599
|
0.9878
|
0.9577
|
0.9765
|
0.9827
|
|
Multivariate two dimensional singular spectrum analsysis [31] |
0.2124 |
0.4512 |
0.4675 |
0.3345 |
0.4954 |
0.4658 |
0.3654 |
0.3789 |
0.5614 |
|
Three dimensional real valued singular spectrum analysis [31] |
0.8282
|
0.8789
|
0.951
|
0.9194
|
0.9446
|
0.9793
|
0.9177
|
0.9833
|
0.9906
|
|
Principal component analysis based method[7] |
0.6972
|
0.7393
|
0.7906
|
0.7256
|
0.7923
|
0.8439
|
0.9095
|
0.93
|
0.8439
|
|
Median filtering based method [4][9] |
0.8974
|
0.9291
|
0.9562
|
0.8974
|
0.9245
|
0.9424
|
0.9581
|
0.9824
|
0.9803
|
|
Tucker decomposition based method [33] |
0.4536
|
0.5748
|
0.6566
|
0.8207
|
0.897
|
0.9366
|
0.9468
|
0.959
|
0.9803
|
|
HybridSN [34] |
0.6321 |
0.8035 |
0.9811 |
0.9451 |
0.9810 |
0.9821 |
0.9231 |
0.9754 |
0.9905 |
Point 9: There are confusing statements in Section 4.1. It is confusing to say “Three hyperspectral images acquired…” I guess the authors are referring to three different categories of hyperspectral image databases or corpora (NASA, Indian Pines, Botswana). Each image data corpus should be described in clear terms in tandem. The authors should provide references or web links to the databases.
Response 9:Agree.
We have deleted the confusing statements and the web links to the databases has been provided in 2.3:KSC(http://aviris.jpl.nasa.gov/),Indian Pines(http://aviris.jpl.nasa.gov/) and Botswana(http://www.ehu.eus/ccwintco/index.php?title=Hyperspectral_Remote_Sensing_Scenes).
Point 10 The last paragraph in section 4.2 is not appropriate in the method section and should be deleted.
Response 10. Agree.
The last paragraph in section 4.2 has been deleted.
Point 11 The first paragraph in section 4.3 is not appropriate for the section theme and should be deleted.
Response 11. Agree.
The first paragraph in section 4.3 has been deleted.
Point 12 The symbols for penalty and gamma parameters should not be written because authors are assuming that readers are familiar with their implementation tool in section 4.3. Simple statement such as “the penalty and gamma parameters were tuned …” is sufficient. The authors can further explain the physical interpretation of the parameters, but it is not important to list their symbols.
Response 12 Agree.
The symbols have been removed.
Point 13 The authors should provide references to other pre-processing methods that were compared quantitatively in tables 1 to 4.
Response 13: Agree.
The references has been provided in tables.
Point 14: The kind of artefacts in the hyperspectral images that were enhanced by the proposed method were not explained in the manuscript.
Response 14: Partially agree.
We may misunderstand your comments. The term enhancement does neither for improving the visual quality nor suppressing the artifacts.
Point 15: Authors should define all acronyms where they were first used. They should not assume that readers are familiar with them, for instance NASA.
Response 15: Agree. The acronyms has been predefined in the paper.
Dear Reviewer1, Please see the attachment for the full paper.

Reviewer 2 Report
The manuscript is well-written and looks convincing at first glance. The results are interesting.
However, there are several questions to clarify important points.
- First, I see that the investigated data are 3D. There are different versions of nD-SSA including 3D-SSA, M 2D-SSA and so on (all of them are particular cases of Shaped nD-SSA, where a window of some shape moves through an object of some shape and dimension). Therefore, there are many options for comparison. Probably, one of the relevant references is the book https://www.springer.com/us/book/9783662573785, where these multidimensional versions are described together with their implementation in R. Did the authors consider these versions, e.g. 3D-SSA, where the window is 3D, or M 2D-SSA, when the window is 2D and moves through several 2D slices?
- For some kind of signals, there is the theoretic approach to the comparison of different SSA version through signal ranks. For example, for analysis of two noisy sinusoidal signals, x and y, can be used: SSA (for analysing them separately), Multichannel SSA (x,y) and Complex SSA (time series in the form x+iy). The choice of Complex SSA is appropriate for a specific form of time series with signals consisting of sums of imaginary exponentials, since their ranks equal 1, that is, they have simple structure from the viewpoint of Complex SSA. Since Quaternion SSA can be considered as the extension of Complex SSA from complex-valued series to quaternion-valued series, the question is: did the authors consider the theory of Quaternion SSA (if it exists) to choose this version among multidimensional SSA versions? Why not use e.g. 3D-SSA with a small window size in the spectral dimension?
- The choice of window sizes in SSA-related methods is important. Did you try other window sizes? In general, for different kinds of SSA different window sizes can provide better accuracy.
- The important question is about the validation of the method. The rule of testing is not to use the test point in the algorithm construction/parameter choice to avoid data leakage (the known problem in machine learning). As I understand, QSSA is applied to the whole set of pixels including the test ones. What do the authors think about possible data linkage?
- I do not consider the superiority of the suggested method among the existing methods as a necessary condition. However, it would be useful to give some reference accuracies, e.g. from https://arxiv.org/pdf/1902.06701v3.pdf .
- Links to the considered data would be helpful.
Author Response
Dear reviewer 2, Please see the attachment for the full paper.
Response to Reviewer 2 Comments
Point 1: First, I see that the investigated data are 3D. There are different versions of nD-SSA including 3D-SSA, M 2D-SSA and so on (all of them are particular cases of Shaped nD-SSA, where a window of some shape moves through an object of some shape and dimension). Therefore, there are many options for comparison. Probably, one of the relevant references is the book https://www.springer.com/us/book/9783662573785, where these multidimensional versions are described together with their implementation in R. Did the authors consider these versions, e.g. 3D-SSA, where the window is 3D, or M 2D-SSA, when the window is 2D and moves through several 2D slices?
Response 1: Agree. Thanks for the advices of the comparison of the 3D-SSA and the M2DSSA. The comparison results has been provided in the table 3-7 and section 4. And the introduction of the 3D-SSA and the MSSA are shown in the third paragraph of section 3.1.
Since the investigated data is 3D, there is the version of three dimensional singular spectrum analysis(3D-SSA)[30] [31], where a 3D window moves through the whole data. The flow of the 3D-SSA is as follow: First, for each location of the window, unfolding each window into vectors to obtain the so called trajectory matrix. Then the SVD, the reconstruction stage and the grouping stage are operated on the trajectory matrix to gain the components of the 3D data. Here, for comparison, the window sizes of 3DSSA are chosen as . Besides, there is another SSA technique in SSA family, called multi-variate two dimensional singular spectrum analysis(M2DSSA), which is also used for processing the data with 3D structure. The difference between the M2DSSA and the 2DSSA is that the M2DSSA is able to processing the data with more channels of 2D matrices[31]. Therefore the M2DSSA is also compared to the two dimensional real valued singular spectrum analysis based method, with the window size of and equals to the total number of frequency bands of the hyperspectral images . For having a more fair comparison among the 2Ddssa, M2DSSA, 3DSSA, and the two dimensional quaternion valued singular spectrum analysis, the method for selecting the SSA components is the same as that for selecting the two dimensional quaternion valued singular spectrum analysis components. That is, the SSA components of different SSA techniques are selected in such a way that the ratio of the interclass separation to the intraclass separation of the pixel vectors is maximized. Then, the selected singular spectrum analysis components are summed up together to construct the hyperspectral image. Finally, the pixel vectors of the reconstructed hyperspectral image are used for performing the object recognition.
Table 3. The classification accuracy based on the KSC image.
|
Different preprocessing methods with different parameters |
The overall classification accuracies of different methods at different ratios of the total number of the pixel vectors in the training set to that in the overall image |
|||||||
|
Ratio of the total number of the pixel vectors in the training set to that in the overall image=0.05 |
Ratio of the total number of the pixel vectors in the training set to that in the overall image=0.1 |
Ratio of the total number of the pixel vectors in the training set to that in the overall image=0.2 |
||||||
|
Two dimensional quaternion valued singular spectrum analysis based method |
The two dimensional quaternion valued singular spectrum analysis components are selected such that the ratio of the interclass separation to the intraclass separation of the pixel vectors is maximized. |
90.9604 (Score=8) |
94.8466 (Score=8) |
98.2519 (Score=8) |
||||
|
Two dimensional real valued singular spectrum analysis based method[4] |
The two dimensional real valued singular spectrum analysis components are selected such that the ratio of the interclass separation to the intraclass separation of the pixel vectors is maximized. |
86.2793 (Score=6) |
90.0341 (Score=6) |
93.8937 (Score=5) |
||||
|
Multivariate two dimensional singular spectrum analysis[31] |
The two dimensional real valued singular spectrum analysis components are selected such that the ratio of the interclass separation to the intraclass separation of the pixel vectors is maximized. |
57.0420 (Score=2) |
59.0716 (Score=1) |
65.4454 (Score=1) |
||||
|
Three dimensional real valued singular spectrum analysis[31] |
The two dimensional quaternion valued singular spectrum analysis components are selected such that the ratio of the interclass separation to the intraclass separation of the pixel vectors is maximized. |
89.3261 (Score=7) |
92.3765 (Score=7) |
97.2941 (Score=6) |
||||
|
Principal component analysis based method[7] |
The lengths of the pixel vectors are reduced by |
20% |
70.1977 |
72.8066 |
74.1858 |
|||
|
40% |
75.2825 |
78.5562 |
80.9387 |
|||||
|
80% |
79.7417 (Score=4) |
82.0486 (Score=4) |
85.1054 (Score=3) |
|||||
|
Median filtering based method[4][9] |
The window sizes are |
92.194 |
94.5132 |
97.8259 |
||||
|
93.3188 |
93.0229 |
97.3141 |
||||||
|
93.5835 (Score=9) |
95.0383 (Score=9) |
97.9873 (Score=7) |
||||||
|
90.7869 |
92.3118 |
97.1185 |
||||||
|
Tucker decomposition based method[33] |
The percentages of the P, Q and R are |
10% |
50.1211 (Score=1) |
62.0954 (Score=2) |
68.6782 (Score=2) |
|||
|
20% |
44.8951 |
53.0451 |
61.5182 |
|||||
|
40% |
37.6312 |
43.2069 |
48.3238 |
|||||
|
80% |
45.3793 |
51.2138 |
59.2912 |
|||||
|
HybridSN[34] |
|
|
69.6136 (Score=3) |
79.8933 (Score=3) |
98.3689 (Score=9) |
|||
|
Without the preprocessing (baseline) |
|
84.6651 (Score=5) |
89.5230 (Score=5) |
92.2653 (Score=4) |
||||
Table 4. The classification accuracy based on the Indian image.
|
Different preprocessing methods with different parameters |
The overall classification accuracies of different methods at different ratios of the total number of the pixel vectors in the training set to that in the overall image |
||||
|
Ratio of the total number of the pixel vectors in the training set to that in the overall image=0.05 |
Ratio of the total number of the pixel vectors in the training set to that in the overall image=0.1 |
Ratio of the total number of the pixel vectors in the training set to that in the overall image=0.2 |
|||
|
Two dimensional quaternion valued singular spectrum analysis based method |
The two dimensional quaternion valued singular spectrum analysis components are selected such that the ratio of the interclass separation to the intraclass separation of the pixel vectors is maximized. |
94.6157 (Score=9) |
97.3960 (Score=8) |
98.3371 (Score=7) |
|
|
Two dimensional real valued singular spectrum analysis based method[4] |
The two dimensional real valued singular spectrum analysis components are selected such that the ratio of the interclass separation to the intraclass separation of the pixel vectors is maximized. |
90.8834 (Score=6) |
95.2775 (Score=7) |
98.4984 (Score=9) |
|
|
Multivariate two dimensional singular spectrum analysis[31] |
The two dimensional real valued singular spectrum analysis components are selected such that the ratio of the interclass separation to the intraclass separation of the pixel vectors is maximized. |
49.9111 (Score=1) |
50.8882 (Score=1) |
51.0549 (Score=1) |
|
|
Three dimensional real valued singular spectrum analysis[31] |
The two dimensional quaternion valued singular spectrum analysis components are selected such that the ratio of the interclass separation to the intraclass separation of the pixel vectors is maximized. |
91.8453 (Score=7) |
94.3617 (Score=6) |
97.6297 (Score=6) |
|
|
Principal component analysis based method[7] |
The lengths of the pixel vectors are reduced by |
20% |
73.4135 |
78.2522 |
81.1740 |
|
40% |
73.9676 |
79.7859 (Score=2) |
83.4202 |
||
|
80% |
75.4522 (Score=3) |
79.5763 |
85.1328 (Score=2) |
||
|
Median filtering based method[4][9] |
The window sizes are |
812546 |
88.3041 |
90.9531 |
|
|
85.2274 |
89.6502 |
93.5468 |
|||
|
87.7888 (Score=5) |
92.5742 (Score=5) |
93.1745 |
|||
|
86.5029 |
90.4667 |
94.1549 (Score=5) |
|||
|
Tucker decomposition based method[33] |
The percentages of the P, Q and R are |
10% |
82.7182 (Score=4) |
88.3372 (Score=4) |
92.5540 |
|
20% |
81.3696 |
87.2228 |
92.7277 (Score=4) |
||
|
40% |
78.2122 |
85.1263 |
87.6768 |
||
|
80% |
75.9435 |
81.4190 |
84.6860 |
||
|
HybridSN[34] |
|
93.5503 (Score=8) |
98.1896 (Score=9) |
98.3414 (Score=8) |
|
|
Without the preprocessing (baseline) |
|
75.0742 (Score=2) |
81.3721 (Score=3) |
85.6658 (Score=3) |
|
Table 5. The classification accuracy based on the Botswana image.
|
Different preprocessing methods with different parameters |
The overall classification accuracies of different methods at different ratios of the total number of the pixel vectors in the training set to that in the overall image |
|||||
|
Ratio of the total number of the pixel vectors in the training set to that in the overall image=0.05 |
Ratio of the total number of the pixel vectors in the training set to that in the overall image=0.1 |
Ratio of the total number of the pixel vectors in the training set to that in the overall image=0.2 |
||||
|
Two dimensional quaternion valued singular spectrum analysis based method |
The two dimensional quaternion valued singular spectrum analysis components are selected such that the ratio of the interclass separation to the intraclass separation of the pixel vectors is maximized. |
97.5097 (Score=9) |
99.0106 (Score=9) |
99.6933 (Score=9) |
||
|
Two dimensional real valued singular spectrum analysis based method[9] |
The two dimensional real valued singular spectrum analysis components are selected such that the ratio of the interclass separation to the intraclass separation of the pixel vectors is maximized. |
96.2807 (Score=8) |
98.0894 (Score=6) |
98.4279 (Score=6) |
||
|
Multivariate two dimensional singular spectrum analysis[31] |
The two dimensional real valued singular spectrum analysis components are selected such that the ratio of the interclass separation to the intraclass separation of the pixel vectors is maximized. |
53.2018 (Score=1) |
58.8195 (Score=1) |
62.1933 (Score=1) |
||
|
Three dimensional real valued singular spectrum analysis[31] |
The two dimensional quaternion valued singular spectrum analysis components are selected such that the ratio of the interclass separation to the intraclass separation of the pixel vectors is maximized. |
94.5019 (Score=6) |
98.2600 (Score=7) |
99.0798 (Score=7) |
||
|
Principal component analysis based method[7] |
The lengths of the pixel vectors are reduced by |
20% |
88.5511 |
91.7434 |
94.2868 |
|
|
40% |
88.8745 |
92.6987 (Score=3) |
94.4018 |
|||
|
80% |
90.2975 (Score=3) |
92.5623 |
94.7853 (Score=3) |
|||
|
Median filtering based method[4][9] |
The window sizes are |
92.0440 |
95.8376 |
97.8528 (Score=4) |
||
|
96.0220 |
97.6117 |
96.1123 |
||||
|
96.2807 (Score=7) |
98.3623 (Score=8) |
94.3540 |
||||
|
93.7904 |
95.7352 |
95.6848 |
||||
|
Tucker decomposition based method[33] |
The percentages of the P, Q and R are |
10% |
89.9094 |
95.3599 |
95.4371 |
|
|
20% |
90.3946 |
95.5647 (Score=4) |
98.0445 (Score=5) |
|||
|
40% |
94.1462 (Score=5) |
95.1552 |
97.3160 |
|||
|
80% |
89.7801 |
93.0740 |
96.1273 |
|||
|
HybridSN [34] |
|
|
92.5145 (Score=4) |
97.9480 (Score=5) |
99.2304 (Score=8) |
|
|
Without the preprocessing (baseline) |
|
85.9314 (Score=2) |
91.8458 (Score=2) |
93.4049 (Score=2) |
||
Table 6. The average scores of different methods for different images.
|
Method |
The total scores of different methods for different images |
The average total scores |
|||
|
|
KSC |
Indian |
Botswana |
|
|
|
Two dimensional quaternion valued singular spectrum analysis based method |
24 |
24 |
27 |
25 |
|
|
Two dimensional real valued singular spectrum analysis based method [9] |
17 |
22 |
20 |
19.6667 |
|
|
Multivariate two dimensional singular spectrum analsysis[31] |
4 |
3 |
3 |
3.3333 |
|
|
Three dimensional real valued singular spectrum analysis[31] |
20 |
19 |
20 |
19.6667 |
|
|
Principal component analysis based method[7] |
11 |
7 |
9 |
9 |
|
|
Median filtering based method[4][9] |
25 |
15 |
19 |
19.6667 |
|
|
Tucker decomposition based method[33] |
5 |
12 |
14 |
10.3333 |
|
|
HybridSN [34] |
15 |
25 |
17 |
19 |
|
|
Without preprocessing |
14 |
8 |
6 |
9.3333 |
|
Point 2: For some kind of signals, there is the theoretic approach to the comparison of different SSA version through signal ranks. For example, for analysis of two noisy sinusoidal signals, x and y, can be used: SSA (for analysing them separately), Multichannel SSA (x,y) and Complex SSA (time series in the form x+iy). The choice of Complex SSA is appropriate for a specific form of time series with signals consisting of sums of imaginary exponentials, since their ranks equal 1, that is, they have simple structure from the viewpoint of Complex SSA. Since Quaternion SSA can be considered as the extension of Complex SSA from complex-valued series to quaternion-valued series, the question is: did the authors consider the theory of Quaternion SSA (if it exists) to choose this version among multidimensional SSA versions? Why not use e.g. 3D-SSA with a small window size in the spectral dimension?
Response 2: Many thanks for the suggestion, For the analysis of a three dimensional cube, M2D-SSA, 3D-SSA and the two dimensional quaternion valued singular spectrum analysis can be used.
On the one hand, M2D-SSA and 3D-SSA process the whole 3D cube which may lose the local information among frequency bands. On the other hand, the two dimensional quaternion valued singular spectrum analysis is appropriate for a specific form of the four-channel metrix consisting of sums of , , imaginary parts and real part, since their ranks equal to the row or the column of the matrix, that is, they have simple structure from the viewpoint of two dimensional quaternion SSA.
Point 3: The choice of window sizes in SSA-related methods is important. Did you try other window sizes? In general, for different kinds of SSA different window sizes can provide better accuracy.
Response 3: Agree.
Here is reason why we choose such a window. And we provide the comparison between different window sizes in Table1. Since the selection of the window size is an important factor for the representation of the components, it will influence the results of the experiments. For the singular spectrum analysis, for the purpose of extracting the local relationship between pixels in window, we may choose a smaller window when the pixels are with closer relationship to each other in the smaller domain of a image. In a hyperspectral image, the pixels in one class are with closer relationship and are always gathered in one place in the image. What’s more, the larger the window size we select, the larger of the computational power we cost. Therefore, In this paper, the window size are selected smaller. For choosing the window size, we compare the simulation results of the method with different size of window in different dataset(KSC, Indian pines and Botswana), shown as Table 1(the second paragraph in section 2.2.1):
|
Method |
Window size |
The average classification accuracies of different methods at different ratios of the total number of the pixel vectors in the training set to that in the different dataset |
||||||||
|
KSC |
Indian |
Botswana |
||||||||
|
0.05 |
0.1 |
0.2 |
0.05 |
0.1 |
0.2 |
0.05 |
0.1 |
0.2 |
||
|
Two dimensional quaternion valued singular spectrum analysis based method |
90.0323 |
93.0366 |
96.4320 |
91.3853 |
95.5313 |
97.0340 |
96.8952 |
98.9765 |
99.0699 |
|
|
90.9604
|
94.8466
|
98.2519
|
94.6157 |
97.3960
|
98.3371
|
97.5097
|
99.0106
|
99.6933
|
||
Table 1. The comparison of average classification accuracies for different size of window
Point 4: The important question is about the validation of the method. The rule of testing is not to use the test point in the algorithm construction/parameter choice to avoid data leakage (the known problem in machine learning). As I understand, QSSA is applied to the whole set of pixels including the test ones. What do the authors think about possible data linkage?
Response 4 : In our paper, the QSSA is regarded as the decomposition of the hyperspectral image into the QSSA components. In this step, all the pixel vectors in the hyperspectral image have performed this procedure. On the other hand, the selection of the QSSA components is regarded as a machine learning operator. Here, only the pixel vectors in the training set are employed to learn the relationship between the pixel vectors and the selection coefficients. After performing the training, all the pixel vectors perform the classification based on the learned characteristics.
Point 5: I do not consider the superiority of the suggested method among the existing methods as a necessary condition. However, it would be useful to give some reference accuracies, e.g. from https://arxiv.org/pdf/1902.06701v3.pdf .
Response 5:
Agree. So we have compared the results between the paper(https://arxiv.org/pdf/1902.06701v3.pdf) and the proposed methods:
Recently, the use of CNN for HSI classification becomes more and more and more popular. Therefore, the proposed method is also compared to the CNN based method, Hybrid Spectral Convolutional Neural Network (Hybrid SN)[33](The last paragraph in section 3.1).
The comparison results is shown above in the Table 3-6
Point 6: Links to the considered data would be helpful.
Response 6: Agree.
We have added the web links to the databases has been provided in 2.3 KSC(http://aviris.jpl.nasa.gov/),Indian Pines(http://aviris.jpl.nasa.gov/) and Botswana(http://www.ehu.eus/ccwintco/index.php?title=Hyperspectral_Remote_Sensing_Scenes).
Dear reviewer 2, Please see the attachment for the full paper.

Round 2
Reviewer 1 Report
The quality of the revised version of the manuscript has improved significantly when compared to the original submission. Most of my comments have been adequately resolved to my satisfaction. The manuscript is hereby recommended for publication.
Reviewer 2 Report
The paper was improved, the authors extended the research to the comparison of a larger set of methods. However, a couple of aspects of the research are still unclear.
1) The authors want to show the advantage of Quaternion SSA and the window size is chosen to obtain better accuracy of QSSA; however, the same approach can be performed for other methods separately (best window sizes for each method).
2) If QSSA is applied to pieces consisting of 4 color plane, then, for comparison, 3D-SSA and M 2D-SSA can be applied to the same datasets (pieces of 4 color planes).
Also, the copy-paste error is in the second column of Table 3-5.
Author Response
Response to Reviewer 2 Comments
Point 1: The paper was improved, the authors extended the research to the comparison of a larger set of methods. However, a couple of aspects of the research are still unclear.
Response 1: Many thanks for your appreciations of our works.
Point 2: The authors want to show the advantage of Quaternion SSA and the window size is chosen to obtain better accuracy of QSSA; however, the same approach can be performed for other methods separately (best window sizes for each method).
Response 2: Agree. The window sizes of other methods are also chosen in the revised version of the manuscript in such a way that the highest accuracy is achieved for that particular method.
Point 3: If QSSA is applied to pieces consisting of 4 color plane, then, for comparison, 3D-SSA and M 2D-SSA can be applied to the same datasets (pieces of 4 color planes).
Response 3: Agree. Four color planes of the hyperspectral image form a group and the color planes in each group are processed individually in the revised version of the manuscript. The results are shown in Table 3 to Table 5.
Point 4: Also, the copy-paste error is in the second column of Table 3-5.
Response 4: Agree. The copy-paste error is fixed in the revised version of the manuscript.